# Oxidative Inactivation of SARS-CoV-2 on Photoactive AgNPs@TiO_2_ Ceramic Tiles

**DOI:** 10.3390/ijms22168836

**Published:** 2021-08-17

**Authors:** Ridha Djellabi, Nicoletta Basilico, Serena Delbue, Sarah D’Alessandro, Silvia Parapini, Giuseppina Cerrato, Enzo Laurenti, Ermelinda Falletta, Claudia Letizia Bianchi

**Affiliations:** 1Department of Chemistry, University of Milan, Via Golgi 19, 20133 Milan, Italy; ermelinda.falletta@unimi.it; 2Department of Biomedical, Surgical and Dental Sciences, University of Milan, Via Carlo Pascal 36, 20133 Milan, Italy; nicoletta.basilico@unimi.it (N.B.); serena.delbue@unimi.it (S.D.); 3Department of Pharmacological and Biomolecular Sciences, University of Milan, Via Carlo Pascal 36, 20133 Milan, Italy; sarah.dalessandro@unimi.it; 4Department of Biomedical Sciences for Health, University of Milan, Via Carlo Pascal 36, 20133 Milan, Italy; silvia.parapini@unimi.it; 5Department of Chemistry, University of Turin, Via Pietro Giuria 7, 10125 Turin, Italy; giuseppina.cerrato@unito.it (G.C.); enzo.laurenti@unito.it (E.L.)

**Keywords:** *SARS-CoV-2*, photocatalytic inactivation, surface transmission, AgNPs@TiO_2_, photoactive tiles

## Abstract

The current *SARS-CoV-2* pandemic causes serious public health, social, and economic issues all over the globe. Surface transmission has been claimed as a possible *SARS-CoV-2* infection route, especially in heavy contaminated environmental surfaces, including hospitals and crowded public places. Herein, we studied the deactivation of *SARS-CoV-2* on photoactive AgNPs@TiO_2_ coated on industrial ceramic tiles under dark, UVA, and LED light irradiations. *SARS-CoV-2* inactivation is effective under any light/dark conditions. The presence of AgNPs has an important key to limit the survival of *SARS-CoV-2* in the dark; moreover, there is a synergistic action when TiO_2_ is decorated with Ag to enhance the virus photocatalytic inactivation even under LED. The radical oxidation was confirmed as the the central mechanism behind *SARS-CoV-2* damage/inactivation by ESR analysis under LED light. Therefore, photoactive AgNPs@TiO_2_ ceramic tiles could be exploited to fight surface infections, especially during viral severe pandemics.

## 1. Introduction

In December 2019, an unknown *SARS-CoV-2* virus was detected in the middle of China (Wuhan) [1], and on March 2020, the World Health Organization (WHO) declared it as a COVID-19 pandemic [2]. COVID-19 pathophysiology in most cases results in acute respiratory distress syndrome (ARDS) and gastrointestinal damage, and it also affects the nervous system, whereas elderly-aged people are more vulnerable to *COVID-19* complications, especially those with a chronic critical illness (CI) [3,4]. To date, many infection cases and high daily mortality are still recording among the global populations, and the WHO worldwide recorded more than 208 million confirmed *COVID-19* cases, including more than 4.3 million deaths in mid-August 2021.

*COVID-19* was characterized by its highly person-to-person infection transmission via several routes [5,6]. Microdroplets can remain longer in the air, which elicits the risk of infections at up to 2 m from the infected person [7]. The indirect transmission through infected objects and environmental surfaces has also been considered if susceptible people touch contaminated objects/surfaces and then transfer the virus to themselves. This transmission might happen in highly viral contaminated places such as infected people’s rooms, clinics, and hospitals [5,8] but even in more common areas such as supermarkets, shops, gyms, restaurants, etc. The *COVID-19* contamination in Wuhan (China) was found to be heavy in intensive care rooms, wherein the highly *COVID-19* contamination was detected on floors and objects (bed, computer mice, trash cans, etc.) [9] and in infected patients’ hospital rooms [10,11]. International health bodies, including the WHO, ask for the implementation of social distancing, hand washing, continuous cleaning of objects and surfaces, and droplet precautions [12,13]. In May 2020, the WHO published a guidance report entitled *Cleaning and disinfection of environmental surfaces in the context of COVID-19* [14], after confirming *COVID-19* transmission through contaminated environmental surfaces [15]. In this report, the WHO suggested using chlorine-based disinfectants in particular for those situations in which the cleaning of surfaces using common disinfectants [16] is inconvenient in emergency cases, or even while some disinfectants are ineffective such as the common chlorhexidine digluconate disinfectant [17]. We have to remember that the over-use of disinfectant sprays and products might lead to severe health effects associated primarily with asthma and respiratory disease [18,19]. 

The lifetime of *SARS-CoV-2* on different objects significantly depends on the nature of materials and environmental characteristics [20]. It was widely reported that *SARS-CoV-2* stays alive mostly on smooth surfaces, such as windows, smooth ceramics, doorknobs, countertops, etc. Unlike other previous coronaviruses, some reports mentioned that *SARS-CoV-2* can survive up to 21 days on environmental surfaces, which highly increases the chance of transmission [21,22].

In the present research, we suggest the employment of photocatalytic self-cleaning surfaces functionalized with silver-decorated TiO_2_ (AgNPs@TiO_2_ tiles) as an eco-technology in areas likely to be exposed to severe viral/bacterial infections such as clinics and hospitals to prevent the surface transmissions of microbial pathogens, including COVID-19. The photocatalytic concept is based on the surface coating of ceramic materials with photoactive compounds that, under light irradiation, produce highly oxidative radical oxygen species (ROSs). In turn, onto a photocatalytic surface, pollutants or pathogenic species undergo continuous degradation processes via ROSs [23,24]. In this specific case, the co-presence of Ag and TiO_2_ allows the final material to possess all the photocatalytic properties given by the photocatalytic coating along with the antibacterial/antiviral property already in the dark, thanks to the Ag action [25,26]. 

In this investigation, we tested the antiviral activity of industrially coated AgNPs@TiO_2_ tiles against the deactivation of *SARS-CoV-2* in both dark and light conditions. The photocatalytic activity was already verified both on the bare powder and the industrially digitally printed ceramic tile, against different toxic air/water compounds such as organic dyes [27], drugs [28], and in abatement processes of NOx [29,30] and phenol [31]. Different bacteria strains were investigated, and both the complete degradation and the absence of the biofilm formation were also confirmed [26,32]. The crucial role of AgNPs@TiO_2_ in pathogenic inactivation is due to the excellent combination of AgNPs species, which is known for its natural antimicrobial activity [33,34], and the synergetic photocatalytic effects of TiO_2_ [35,36].

## 2. Results and Discussion 

### 2.1. Ag-TiO_2_ Tile Characterization 

The HR-TEM image in (Figure 1a) shows that TiO_2_ possesses the typical features of a micrometric titania system, with ordered and roundish particles (exhibiting diameter larger than 100 nm and typical anatase phase [28], on top of which Ag species are evident. The latter species could be found both in metallic form (Ag^0^, as indicated by the FFT analysis as shown in Appendix A) and Ag^2+^ species (in the form of Ag_2_O, as already reported in previously detailed research devoted to this topic [26,29]).

As for the morphology of the Ag@TiO_2_ particles on the surface tile, they have been characterized by HR-SEM investigation (Figure 1b). TiO_2_ particles exhibit sizes > 100 nm range (mean diameter), confirming thus the indications coming from HR-TEM observations (see above), with a quite uniform distribution of TiO_2_. EDX mappings relative to either Ti or Ag (Figure 1d,e) confirm this evidence, indicating a good distribution of Ti and Ag species on the surface of the engineered tile. Then, the tile surface was investigated by XPS to verify the composition of the external layer of the engineered surface: the analysis has been carried out both before and after the digital coating process, and survey spectra are reported in (Figure 1e). Characteristic peaks of Ti_2p_ and Ag_3d_ were detected for the AgNPs@TiO_2_ coated tile [37], as well as Si 2p and 2s peaks due to the silicate present in the ink formulation [38]. 

### 2.2. Antiviral Results 

For comparison, antiviral experiments were performed on glass and AgNPs@TiO_2_ tile surfaces in dark and light irradiation (UVA or LED) with a starting viral load of 1.4 × 10^5^ PFU/cm^2^ (log_10_ 5.146). The antiviral activity for *SARS-CoV-2* was expressed in log_10_ reduction as described (see Section 2.1). Figure 2 shows comparatively the results of *SARS-CoV-2* inactivation under different conditions. Detailed results are shown in Appendix A. 

In the dark, the inactivation of *SARS-CoV-2* was more pronounced on the surface of AgNPs@TiO_2_ tile compared to the glass surface within 4 h. Many reports declared that the lifetime of *SARS-CoV-2* is longer on a smooth surface such as glasses. The chance of *SARS-CoV-2* survival depends on the interface interaction of *SARS-CoV-2* droplets, contact angle humidity, and temperature [39]. Due to the smooth surface of glasses, the lifetime of droplets would be longer compared to ceramics materials. In addition, the Ag-rich tiles might prevent the survival or/and inactivate *SARS-CoV-2* due to the antiviral effect of AgNPs [40]. Jeremiah et al. [41] studied the antiviral activity of AgNPs against *SARS-CoV-2*, and it was found that 10 nm diameter AgNPs showed excellent inhibition of extracellular *SARS-CoV-2*. On the glass surface under UVA, a 1.042 and 1.359 log_10_ reduction was observed within 4 and 7 h, respectively. Under these circumstances and compared with the experiment on glass in the dark, we may confirm that the intensity of UVA radiation at the range of 315–400 nm is not strong enough to deactivate *SARS-CoV-2* effectively. Several reports investigated the inactivation of *SARS-CoV-2* by direct deep UV irradiation [42,43,44]. Heilingloh et al. [45] reported that *SARS-CoV-2* was very susceptible to UVC irradiation, while a low inactivation was found under UVA. UVC is the most common radiation used for the inactivation of viruses due to its germicidal effect wavelength peak, which fits with the absorption of nucleic acids [46,47]. 

In terms of Ag-TiO_2_ tile under UVA, the inactivation of the virus is much higher compared to that on the glass surface. Analysis of *SARS-CoV-2* inactivation indicated that a 1.775 and 2.620 log_10_ reduction (corresponding to 98.3% and 99.7% viral reduction) was obtained at 4 and 7 h of contact time, respectively. To check the performance in the presence of higher UVA radiation, an experiment was carried out under UVA with an intensity of 0.25 mW/cm^2^ with a starting log_10_ PFU/cm^2^ of 3.079, and complete inactivation of *SARS-CoV-2* was recorded within 7 h of irradiation.

To exclude any direct cytotoxicity of the surface wash solution on the host cells, a cytotoxicity assay was performed using broth recovered by the glass or Ag@TiO_2_ surface. As shown in the Appendix A, no cytotoxicity against VERO cells was observed. 

In UV-exposed Ag@TiO_2_ tile, the inactivation of *SARS-CoV-2* is due to the generation of reactive oxygen species (ROSs) due to the photoexcitation of Ag@TiO_2_ coated on the surface of the ceramic tile. The AgNPs and TiO_2_ heterojunction system in the presence of light irradiation can be very powerful for the generation of high yield of ROSs, e.g., ^•^OH radicals as already demonstrated in the *E. Coli* degradation in our previous study [26], in which the oxidative inactivation of *E. Coli* was investigated on the same samples (surface of Ag@TiO_2_ tiles) used in the present research. In that work, we observed that the light irradiation of the photocatalytic engineered surface led to shifting the interfacial potential due to (i) the surface stabilization and (ii) the photoproduction of ROSs on the active surface, bringing about the damage of *E. Coli* species. An oxidative synergistic effect resulting from the most photoproduced ROSs (^•^OH, ^−•^O_2_, and H_2_O_2_), together with the direct inactivation via the positive holes on the valence band of Ag@TiO_2_ are cooperatively responsible for the damage of microbial membranes. Long-lived H_2_O_2_ is known to be very powerful in terms of bacterial/viral inactivation because of its relative stability, which is requested for effective damage of microbial membrane [47,48,49], and this latter could be photocatalytically produced from the dimerization and reduction of ^•^OH and ^−•^O_2_, respectively.

Some research groups using different photocatalysts, including TiO_2_, have reported the photocatalytic abatement of *SARS-CoV-2* through oxidative damage [48,49,50]. Under LED light, the log_10_ reduction values were 1.323 and 2.210 within 7 h on glass and Ag@TiO_2_ tile, respectively, confirming the importance of both light irradiation and the presence of a photocatalytic surface to accelerate the viral inactivation.

The photocatalytic generation of ^•^OH species to confirm the radical oxidation of the *SARS-CoV-2* was checked by ESR analysis under similar solar light with UV light cut-off (400 nm) using DMPO as a trapping agent: the relevant spectra are reported in Figure 3. Unlike bare TiO_2_, the pattern of the typical signals of DMPO-^•^OH adduct [51] was notably detected in Ag@TiO_2_, confirming the high photonic synergism obtained in Ag-TiO_2_. 

Based on the obtained results, two major mechanistic pathways could take place on the surface of Ag@TiO_2_ porcelain-grès tiles toward *S**ARS-CoV-2* inactivation (Figure 4). In the dark, slow direct inactivation of *S**ARS-CoV-2* can take place by AgNPs deposited on the surface. AgNPs can directly destroy the membrane of virus, and also, it is an excellent inhibitor against microbial growth. Under the light, fast deactivation of *S**ARS-CoV-2* is a result of the radical oxidation which is produced continuously on the surface of photoactive tiles under UVA or LED. In real conditions, the yield of the spreading virus would be much lesser than the starting yield of *S**ARS-CoV-2* tested in this study. Therefore, antiviral ceramics would be an excellent option mainly to prevent viral transmission, including *S**ARS-CoV-2*, in highly contaminated environments during the critical viral situation, especially in hospitals [52]. 

## 3. Materials and Methods

### 3.1. Fabrication of Ag-Decorated TiO_2_ Photoactive Tiles 

Firstly, the Ag-decorated TiO_2_ powder was prepared by the impregnation method [25]. Commercial 1077-Kronos with particle size in the 110–130 nm range and surface area of 12 ± 2 m^2^g^−1^ was employed as a TiO_2_ photocatalyst source. AgNPs were synthesized starting from silver nitrate (AgNO_3_, ACS Reagent, Sigma-Aldrich, ≥99%), as Ag precursor, in the presence of KNO_3_ (ACS Reagent, Sigma-Aldrich ≥ 99.0%) and polyvinylpyrrolidone (PVP40, average mol·wt^−1^: 40,000, Sigma-Aldrich). AgNPs ratio in the Ag@TiO_2_ composite is 8%. The coating of tiles was industrially produced by digital printing (Projecta Engineering S.r.l., Fiorano M.se, Italy) on porcelain-grès tiles (IrisCeramica Group—Active R&D production site Castellarano, Italy) and stabilized in a kiln at 680 °C; details of this process were provided in previous works [26,53].

### 3.2. Antiviral Experiments and Calculations 

#### 3.2.1. Photocatalytic Antiviral Tests

Photocatalytic experiments were carried out in the dark and using a mercury UVA lamp (500W, Jelosil, Vomodrone, Italy) at 0.1 and 0.25 mW/m^2^ or a LED (Philips, Germany) at 1000 lux conditions, for 4 and 7 h. Experiments were carried out using glass plates for comparison purposes. Glass was properly chosen as the tile surfaces, which are glazed due to the presence of silicate in the coating formulation and its stabilization at 680 °C. 

#### 3.2.2. Viruses and Cells 

*SARS-CoV-2* was isolated from a nasal-pharyngeal swab positive for *SARS-CoV-2*. The complete nucleotide sequence of the *SARS-CoV-2* isolated strain was deposited at Gen Bank, at NCBI (accession number: MT748758).

VERO (Monkey Kidney Epithelial Cells, clone E6, ATCC CRL-1586™) cells were maintained in DMEM medium (EuroClone, Pero, Italy) supplemented with 10% heat-inactivated fetal calf serum, 2 mM glutamine, 100 units/mL of penicillin, and 100 μg/mL of streptomycin (EuroClone, Pero, Italy). 

#### 3.2.3. Preparation of Test Specimens

Each sample (a flat square of (50 ± 2) mm × (50 ± 2) mm) was sterilized by immersion in ethanol 70%, to eliminate any bacterial contamination.

#### 3.2.4. Test Procedure

ISO 18,061 protocol was chosen to test the samples with modifications. Both glass and Ag@TiO_2_ tile were inoculated with 0.2 mL of virus suspensions (1–5 × 10^6^ Plaque-Forming Unit (PFU)/mL), and the inoculum was covered with a 40 × 40 mm film and incubated for 4 or 7 h at room temperature in the dark or under the selected lighting system (LED or UVA).

At the end of the contact time, 20 mL of neutralizer SCDLP broth were added to the samples and plaque assay was performed, in 6-well plates, testing 4-fold serial dilutions of the recovered SCDLP broth in complete medium. Briefly, the cells monolayer was inoculated for 2 h, with 0.4 mL of the virus suspension, recovered in SCDLP broth; each dilution was tested in duplicate. Then, the inoculum was removed, the cells were washed with PBS, covered with 0.3% agarose dissolved in cell medium, and incubated for 48 h at 37 °C, 5% CO_2_. Cells were fixed with 4% formaldehyde solution (Sigma-Aldrich) and, after agarose removal, stained with methylene blue (Sigma-Aldrich). Plaques were counted, and the infectivity titer of the virus was expressed as PFU/cm^2^. The antiviral activity for *SARS-CoV-2* was expressed in log_10_ reduction (log_10_ PFU/cm^2^ at Time 0 (t_0_)-log_10_ PFU/cm^2^ at the subsequent time points).

At T = 0, immediately after virus inoculum, 20 mL of neutralizer SCDLP broth were added to 3 glass samples, and the residual virus infectivity was revealed by plaque assay.

#### 3.2.5. Cytotoxicity and Cell Sensitivity to Virus

For the cytotoxicity assay, cells were seeded into 96-well plates at a concentration of 1.3×10^4^ cells/well. Twenty mL of neutralizer SCDLP broth were added to 3 glass and 3 photoactive samples, and immediately, 0.1 mL was recovered and added to the cells in triplicate. After 2 h of incubation, the SCDLP broth was replaced with a complete medium, and cells were incubated for 48 h at 37 °C in 5% CO_2_. At the end of incubation, cell viability was measured by MTT (3-[4.5-dimethylthiazol-2-yl]-2.5-diphenyltetrazolium bromide) assay. Twenty µL of MTT solution (5 mg/mL) was added to each well for 3 h. Then, the plates were centrifuged, the supernatants discarded, and the resulting pellets dissolved in 100 µL of lysing buffer consisting of 20% (*w*/*v*) of a solution of SDS (Sigma-Aldrich), 40% of N,N-dimethylformamide (Sigma-Aldrich)) in H_2_O. The absorbance was measured spectrophotometrically at a test wavelength of 550 nm and a reference wavelength of 650 nm, using a Synergy 4 microplate reader (Biotek, GE). 

### 3.3. Materials Characterization

#### 3.3.1. DMPO-^•^OH ESR Analysis

ESR analysis to check the photocatalytic generation of ^•^OH on powdered Kronos 1077 TiO_2_ and the same sample decorated (Ag-TiO_2_) was carried out by using as the irradiating source a solar box (Co. Fo. Megra, Milan, Italy) equipped with a 1500 W Xenon lamp and cut-off filters for wavelengths below 340 nm or 400 nm. Then, 3 mL of a sample suspension (prepared to introduce 100 mg of sample in 100 mL of pure water) were introduced in a quartz cell and irradiated under stirring for 20 min in the presence of 5,5-dimethyl-1-pyrroline-N-oxide (DMPO, 17 mM). ESR spectra were recorded at room temperature using an X-band Bruker-EMX spectrometer equipped with a cylindrical cavity operating at 100 kHz field modulation. Experimental parameters were as follows: microwave frequency 9.86 GHz; microwave power 2.7 mW; modulation amplitude 2 Gauss; conversion time 30.68 ms.

#### 3.3.2. HR-TEM Characterization

HR-TEM images have been obtained employing a Jeol JEM 3010-UHR (Japan) microscope equipped with LaB_6_ filament (potential acceleration 300 kV). Images were digitally acquired using an Ultrascan 1000 camera and processed with Gatan Digital Micrograph program version 3.11.1. Before the analysis, samples were dry dispersed onto Cu grids coated with lacey carbon film.

#### 3.3.3. HR-SEM Characterization

A Field Emission Electron Scanning Microscope (FE-SEM) LEO 1525 ZEISS (Germany) was used to determine the photocatalyst distribution at the ceramic surface. Samples were deposited on conductive carbon adhesive tape and metalized with chromium.

#### 3.3.4. X-ray Photoelectron Spectroscopy

An M-probe apparatus (XPS–M-Probe, Surface Science Instruments, USA) recorded the XPS spectra. The instrument is equipped with a monochromatic AlKα anode and a C1s peak at 284.6 eV was used as internal calibration [54]. The energy scale was calibrated with reference to the 4f_7/2_ level of a freshly evaporated gold sample, which was taken as 84.00 eV and with reference to the 2p_3/2_ level of copper taken as 932.47 ±0.10 eV and to the 3s level of copper (122.39 ± 0.15 eV), respectively. An electron gun at 7 eV was used for the analyses of insulating samples. 

## 4. Conclusions 

In the present report, we showed the antiviral activity of AgNPs-decorated TiO_2_ based photoactive ceramic tiles toward *SARS-CoV-2* under dark, UVA, and LED light irradiations. Compared to control experiments on a glass surface, Ag@TiO_2_ photoactive ceramic tiles showed faster *SARS-CoV-2* inactivation. In dark conditions, the inactivation of *SARS-CoV-2* is several times faster on Ag-TiO_2_ ceramic tile than on glass. In addition, UVA irradiation significantly boosts the inactivation of *SARS-CoV-2* on Ag@TiO_2_ ceramic tile via radical oxidation, which was confirmed by ESR analysis. Total *SARS-CoV-2* inactivation with starting Log PFU/cm^2^ = 3.079 was found using UVA intensity of 0.25 mW/cm^2^. Under UVA at a lower intensity of 0.1 mW/cm^2^, high inactivation rates were recorded as well. The radical oxidative inactivation of *SARS-CoV-2* was also found under LED indoor light irradiation. The results of this investigation showed the potential of self-cleaning photoactive materials toward viral inactivation. Exploitation and extending this sustainable technology in heavy contaminated environmental surfaces may help to reduce viral surface transmission, especially during a strong viral pandemic such as the case of *SARS-CoV-2*. 

## Figures and Tables

**Figure 1 ijms-22-08836-f001:**
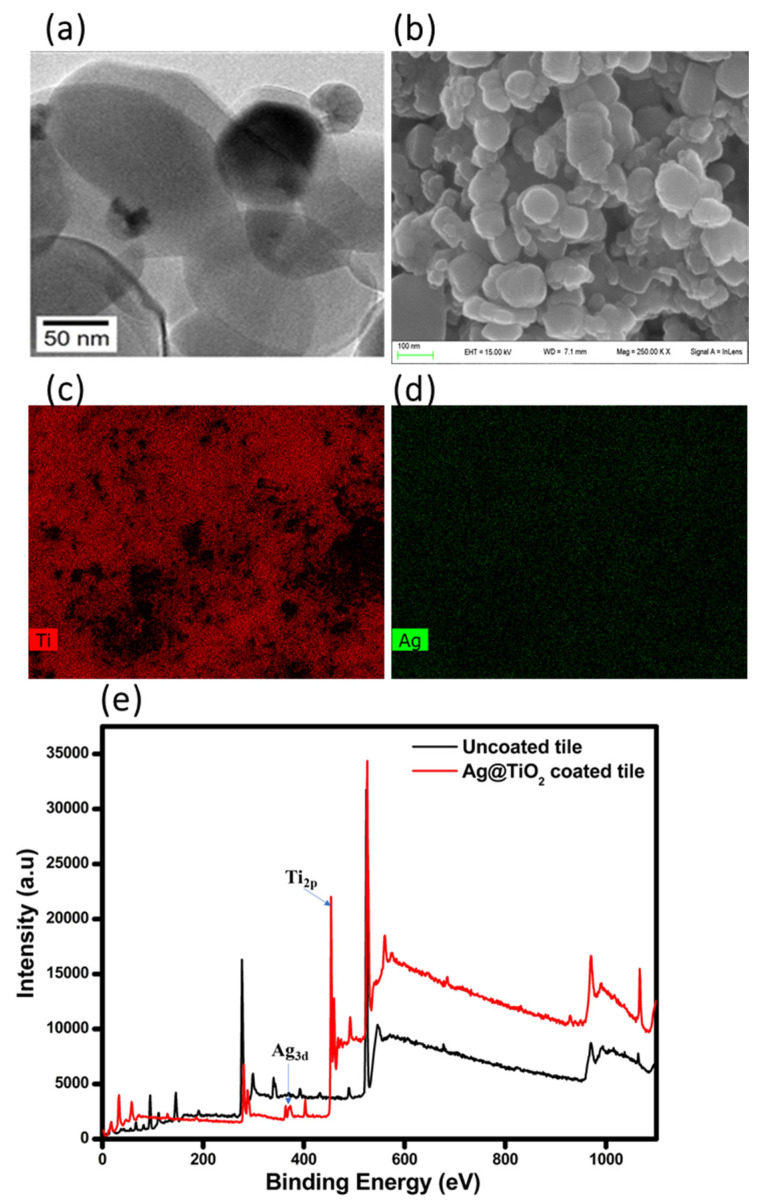
(**a**) HR-TEM images of Ag@TiO_2_ powder. (**b**) SEM images of 8% Ag-decorated TiO_2_ particles. (**c**,**d**) EDX mapping of Ti and Ag species, respectively, on the photocatalytic porcelain-grès tile. (**e**) XPS survey spectra of uncoated and Ag@TiO_2_ coated tiles.

**Figure 2 ijms-22-08836-f002:**
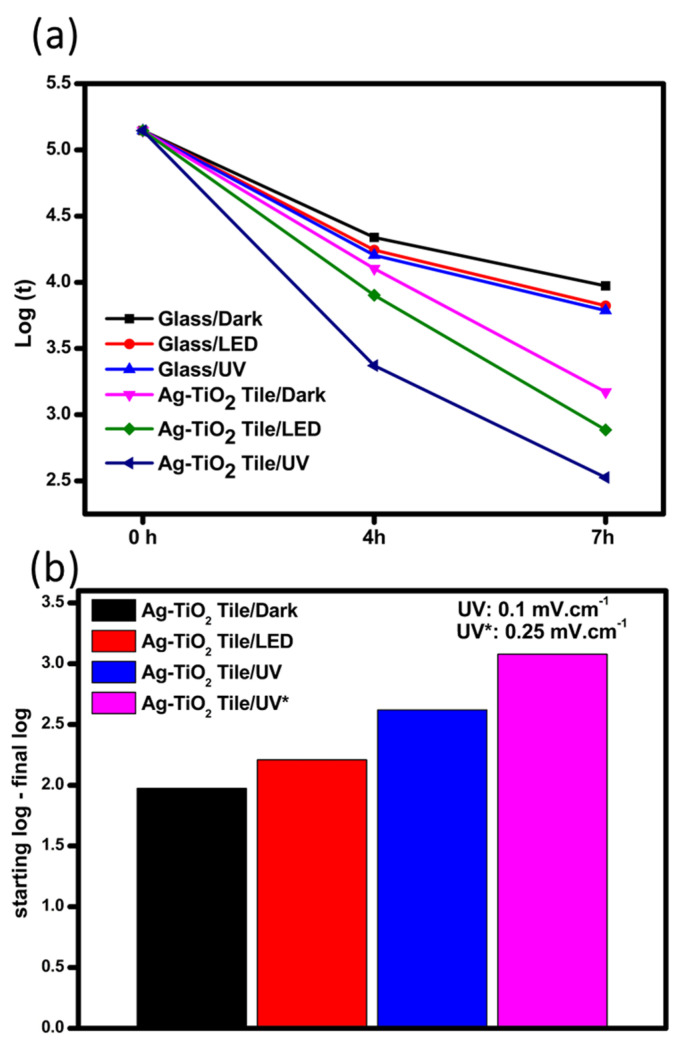
(**a**) Inactivation of *SARS-CoV-2* under different conditions (dark, LED, UVA: 0.1 mW/cm) after 4 and 7 h. (**b**) Selected results under different dark/lighting conditions (LED and UVA: 0.1 and 0.25 mW/cm) after 7 h.

**Figure 3 ijms-22-08836-f003:**
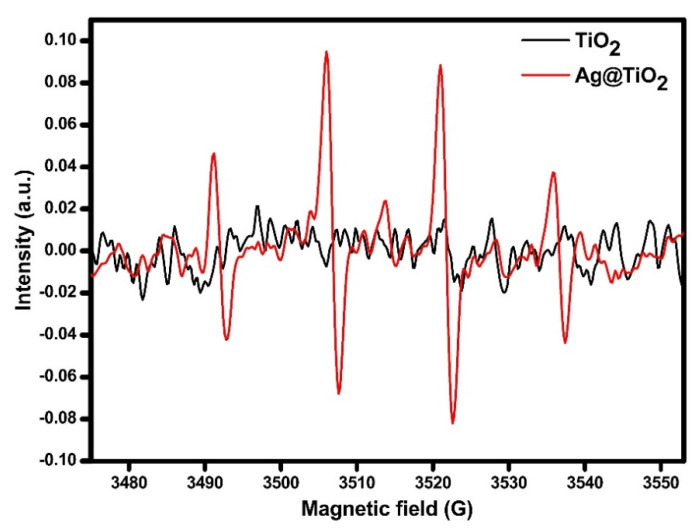
ESR analysis using DMPO as a trapping agent under similar solar light with UV light cut-off (>400 nm).

**Figure 4 ijms-22-08836-f004:**
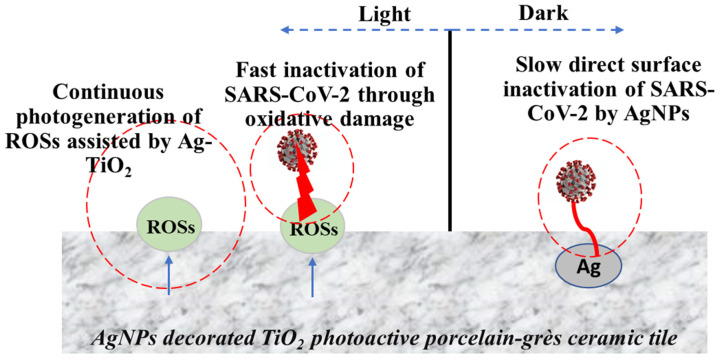
Mechanistic pathways of *SARS-CoV-2* inactivation on the surface of Ag-decorated TiO_2_ porcelain-grès tiles with light irradiations (UVA or LED) or in the dark.

## Data Availability

Not applicable.

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
