# Peer review of "Oxidative Inactivation of SARS-CoV-2 on Photoactive AgNPs@TiO2 Ceramic Tiles"

_ijms, 2021, doi:10.3390/ijms22168836_

Round 1
Reviewer 1 Report
The paper concerns oxidative deactivation of SARS-CoV-2 on photoactive Ag NPs at TiO2 ceramic tiles
The work is innovative and I suggest publication
Author Response
The authors thank the reviewer for the appreciation of the work.
Reviewer 2 Report
The manuscript is a nice piece of work detailing how AgNPs in combination with TiO2 and light irradiation can combine to form an effective antiviral surface suitable for use in highly contaminated areas. The work is scientifically sound and likely of interest to readers of the journal. I have a few suggestions that the authors should address.
The introduction is rather long and could probably be condensed somewhat.
In the antiviral results section (lines 137-149) there is a lengthy description of the results pertaining to inactivation of E. coli from another of the authors papers (ref 31). While this is relevant, it is not appropriate to repeat the results/inactivation mechanism again in this work and this section should be shortened.
The inset and labels in figure 1a are too small to see clearly, the inset may be better as a supplementary figure.
Figure 2b does not, in my opinion add anything to the already very clear results shown in 2a, perhaps a table shown log difference and percentage inactivation after each treatment would be more informative.
The methods section lacks some detail, for example in the XPS section, instrument setting should be included.
The Author contributions/funding/acknowledgements sections are currently incomplete.
Finally, I would recommend a thorough proofread to check the quality of the English used as there are a lot of small errors throughout the manuscript.
Author Response
The introduction is rather long and could probably be condensed somewhat.
Reply: as suggested, the introduction was adequately rewritten and shortened.
In the antiviral results section (lines 137-149), there is a lengthy description of the results pertaining to the inactivation of E. coli from another of the authors papers (ref 31). While this is relevant, it is not appropriate to repeat the results/inactivation mechanism in this work and this section should be shortened.
Reply: thank you for the suggestion. This point was taken into account, and the section was shortened.
The inset and labels in figure 1a are too small to see clearly, the inset may be better as a supplementary figure.
Reply: as suggested, the inset in Figure 1a was removed.
Figure 2b does not, in my opinion, add anything to the already very clear results shown in 2a, perhaps a table shown log difference and percentage inactivation after each treatment would be more informative.
Reply: we do not agree with the referee. Table 1 SI was adequately implemented as requested by the reviewer, but we decided to maintain Fig.2b, too as it reports the inactivity values using two different UVA conditions and this could be extremely interesting for the readers.
The methods section lacks some detail, for example in the XPS section, instrument setting should be included.
Reply: XPS setting details were added.
The Author contributions/funding/acknowledgments sections are currently incomplete.
Reply: The author's contributions, funding, and acknowledgment were added.
Finally, I would recommend a thorough proofread to check the quality of the English used as there are a lot of small errors throughout the manuscript.
Reply: English was carefully checked.